# Comparison of the Antihypertensive Activity of Phenolic Acids

**DOI:** 10.3390/molecules27196185

**Published:** 2022-09-21

**Authors:** Myeongnam Yu, Hyun Joo Kim, Huijin Heo, Minjun Kim, Yesol Jeon, Hana Lee, Junsoo Lee

**Affiliations:** 1Department of Food Science and Biotechnology, Chungbuk National University, Cheongju 28644, Korea; 2Department of Central Area Crop Science, National Institute of Crop Science, Rural Development Administration, Suwon 16613, Korea

**Keywords:** phenolic acids, sinapic acid, ACE inhibition, hypertension, endothelial dysfunction

## Abstract

Phenolic acids, found in cereals, legumes, vegetables, and fruits, have various biological functions. We aimed to compare the antihypertensive potential of different phenolic acids by evaluating their ACE inhibitory activity and cytoprotective capacity in EA.hy 926 endothelial cells. In addition, we explored the mechanism underlying the antihypertensive activity of sinapic acid. Of all the phenolic acids studied, sinapic acid, caffeic acid, coumaric acid, and ferulic acid significantly inhibited ACE activity. Moreover, gallic acid, sinapic acid, and ferulic acid significantly enhanced intracellular NO production. Based on the results of GSH depletion, ROS production, and MDA level analyses, sinapic acid was selected to study the mechanism underlying the antihypertensive effect. Sinapic acid decreases endothelial dysfunction by enhancing the expression of antioxidant-related proteins. Sinapic acid increased phosphorylation of eNOS and Akt in a dose-dependent manner. These findings indicate the potential of sinapic acid as a treatment for hypertension.

## 1. Introduction

Cardiovascular diseases (CVDs), including coronary artery disease, atherosclerosis, and hypertension, are a group of diseases that acutely threaten human health [1]. Endothelial dysfunction, a hallmark of hypertension, can be caused by oxidative stress. Nitric oxide (NO) is a crucial mediator of endothelium-dependent relaxation in blood pressure regulation [2]. Increased radicals rapidly react with NO, resulting in altered NO bioavailability and impaired endothelial relaxation [3]. Reactive oxygen species (ROS) affect the structure and function of vascular media. Vascular remodeling by ROS leads to enhanced medial thickness [4]. Therefore, preventing oxidation and ROS generation may help improve hypertension-related diseases [5]. Nuclear factor-E2-related factor 2 (Nrf2), an antioxidant transcription factor important in CVD resistance [6], is highly sensitive to oxidative damage. Nrf2 promotes the transcription of antioxidant genes, including heme oxygenase-1 (HO-1), NADPH quinone oxidoreductase (NQO-1), and glutamate-cysteine ligase catalytic subunit (GCLC). A previous study demonstrated that accumulation of Nrf2 in the nucleus and activation of protein kinase B (Akt) accompanied HO-1 and NQO-1 expression [7]. In the endothelium, many growth factors and hormones act as agonists to induce the activation of Akt and phosphorylation of endothelial nitric oxide synthase (eNOS), which increases NO production [8]. As various endothelial signaling pathways converge on Akt, it may be an ideal target protein for eNOS responses [9]. Therefore, the Akt/eNOS and Nrf2 signaling pathways are crucial checkpoints for the induction of phase II enzymes and treatment of endothelial dysfunction.

Phenolic compounds are the most abundant phytochemicals in plant-based foods. Phenolic acids are a major class of phenolic compounds that can suppress ROS, thus reducing oxidative stress to biomolecules within cells [10]. Phenolic acids exert various biological activities, including antioxidant, anticancer, antidiabetic, anti-inflammatory, and antihypertension [11]. The ameliorative effect of phenolic acids on chronic diseases may be due to their high antioxidative potential [12]. Gallic acid suppressed hypertension in L-NAME-treated mice and spontaneously hypertensive rats [13,14]. Ferulic acid reduced oxidative injury by increasing the bioavailability of NO in arterial vasculature [15]. Moreover, chlorogenic acid and caffeic acid lowered blood pressure and decreased the properties of enzymes associated with the pathogenesis of hypertension [16]. However, information on the comparative efficacy of phenolic acids in modulating endothelial dysfunction and hypertension is limited. This study aimed to compare the inhibitory effect of phenolic acids on endothelial dysfunction against the oxidative damages in EA.hy 926 endothelial cells.

## 2. Materials and Methods

### 2.1. Chemicals

Griess reagent, benzoic acid, hydroxybenzoic acid, caffeic acid, cinnamic acid, coumaric acid, ferulic acid, gallic acid, protocatechuic acid, sinapic acid, syringic acid, vanillic acid, quercetin, angiotensin-converting enzyme (ACE), captopril, hydrogen peroxide (H_2_O_2_), and diacetyldichlorofluorescein were purchased from Sigma-Aldrich (St. Louis, MO, USA). Antibodies against p-eNOS, eNOS, p-Akt, Akt, Nrf-2, NQO-1, PCNA, HO-1, GCLC, and β-actin were obtained from Cell Signaling Technology (Beverly, MA, USA). Dulbecco’s modified Eagle’s medium (DMEM), fetal bovine serum (FBS), and penicillinstreptomycin were purchased from Hyclone (General Electric Healthcare Life Sciences, Mississauga, Canada). The structure of phenolic acids is indicated in Table 1.

### 2.2. ACE Inhibitory Activity Assay

ACE inhibitory activity of phenolic acids was determined according to the method reported by Cushman and Cheung (1971), using ACE (0.1 U/mL) and hippuryl-His-Leu (5 mM) [17]. The absorbance was measured at 228 nm using a spectrophotometer (BioTek, Inc., Winooski, VT, USA). Captopril was used as a positive control. The inhibition rate was calculated using the following formula.
ACE inhibition rate (%) = (OD_control_ − OD_sample_)/OD_control_ × 100

### 2.3. Cell Culture and Sample Treatment

EA.hy 926 cells were incubated in DMEM supplemented with 10% FBS at 37 °C in humidified air with 5% CO_2_. Endothelial cells were seeded at a density of 6 × 10^5^ cells/mL in a 96-well plate. The cells were pre-treated with a serum-free medium containing 50 μM phenolic acids for 1 h and then exposed to 600 μM H_2_O_2_ with phenolic acids for 24 h. Cell cytotoxicity was determined using a thiazolyl blue tetrazolium bromide reagent.

### 2.4. Measurement of Intracellular ROS, GSH, Malondialdehyde, and NO Levels

Endothelial cells were treated with 50 μM phenolic acid and 600 μM H_2_O_2_. Next, the cells were washed with PBS and stained with 25 μM diacetyldichlorofluorescein. The fluorescence intensity was analyzed. Glutathione and malondialdehyde levels were measured using the DTNB-GSSG reductase recycling and TBARS assays, respectively. Nitric oxide levels were measured using Griess reagent.

### 2.5. Western Blot Analysis

The EA.hy 926 endothelial cells were cultured in a 6-well plate with or without sinapic acid at a density of 6 × 10^5^ cells/mL. Total proteins and nuclear proteins were extracted using the Pro-Prep^TM^ protein extraction solution (iNtRON Biotechnology, Seongnam, Korea) and NE-PER^®^ nuclear and cytoplasmic extraction reagents (Thermo Fisher Scientific, Inc., Worcester, MA, USA), respectively. Membranes were incubated with primary and secondary antibodies (1:2000 dilution for β-actin; 1:1000 dilution for p-eNOS, eNOS, p-Akt, Akt, NQO-1, PCNA, HO-1, GCLC, anti-mouse, and anti-rabbit; 1:500 dilution for Nrf-2). The bands were visualized using X-ray film.

### 2.6. Statistical Analysis

Data were analyzed using Duncan’s multiple comparison test and Tukey’s post hoc test using SAS (version 8.1; SAS Institute, Cary, NC, USA) and GraphPad Prism software (version 5; GraphPad Software Inc., La Jolla, CA, USA).

## 3. Results and Discussion

### 3.1. Effect of Phenolic Acids on ACE Inhibition and NO Production

ACE plays a crucial role in regulating blood pressure [18]. Many synthetic ACE inhibitors are currently being used for the treatment of hypertension. However, these drugs may cause adverse effects. Most natural compounds are safe and do not cause adverse effects. A previous study reported that plant phenolics have the potential to inhibit ACE in vitro [19]. Zhang et al. (2018) demonstrated the ACE inhibition effect of phenolic extracts and fractions derived from lentils, black soybean, and black turtle bean [20]. To confirm the antihypertensive effect of phenolic acids, we measured the ACE inhibitory activity. As shown in Figure 1, among the selected phenolic acids, sinapic acid showed the highest ACE inhibition rate (89%), followed by caffeic acid (78%). In this study, we used the EA.hy 926 endothelial cell line to evaluate the effect of phenolic acids on NO production. Treatment with phenolic acids (50 μM) did not affect the cytotoxicity of endothelial cells (Figure 2A). Reduced NO levels contribute to hypertension and endothelial dysfunction. NO plays an essential role in the vasorelaxation of large arteries [21]. We found that treatment with gallic acid, sinapic acid, and ferulic acid significantly increased NO production by 85.1, 50.5, and 31.9%, respectively, compared with that in the control group cells (Figure 2B). These results indicate that phenolic acids may improve endothelial dysfunction, consequently regulating blood pressure.

### 3.2. Cytoprotective Effect of Phenolic Acids against Hydrogen Peroxide

Excessive ROS levels lead to endothelial dysfunction and elevated blood pressure [22]. MDA, a marker of oxidative damage, can cause an abnormal physiological state in the body [23]. GSH, an active peptide with good antioxidant activity, can modulate oxidative balance and suppress oxidative damage [24]. In this study, we investigated the cytoprotective effects of phenolic acids on H_2_O_2_-induced oxidative stress in EA.hy 926 endothelial cells. Treatment with H_2_O_2_ (600 μM) decreased cell viability by 24.8%. However, treatments by caffeic, ferulic, gallic, and sinapic acid markedly increased the cell viability by 43.4, 43.6, 35.5, and 39.1%, respectively, compared to H_2_O_2_-induced cells (Figure 3A). To examine whether phenolic acids protect endothelial cells against oxidative damage, we measured ROS, GSH, and MDA levels (Figure 3B–D). Sinapic acid markedly reduced ROS generation by 44.1% compared to that in H_2_O_2_-treated cells. Caffeic acid, cinnamic acid, coumaric acid, ferulic acid, gallic acid, sinapic acid, and syringic acid significantly enhanced the GSH levels. Our findings show that H_2_O_2_ treatment increased ROS levels and decreased intracellular GSH levels, whereas treatment with phenolic acids significantly reduced oxidative damage-induced ROS production and GSH depletion. In addition, we investigated the effect of phenolic acids on oxidative stress-induced lipid peroxidation in EA.hy 926 cells. Among the phenolic acids, sinapic acid showed the strongest inhibitory effect on lipid peroxidation. Lee and Lee (2021) reported that protocatechuic acid and gallic acid significantly decreased ROS levels, thereby regulating insulin resistance [25]. Caffeic acid and chlorogenic acid decreased blood pressure in hypertensive rats by increasing GSH and reducing MDA levels [16]. Taken together, these results suggest that sinapic acid plays a crucial role in the protection of endothelial cells by regulating ROS, MDA, and GSH levels.

### 3.3. Effects of Sinapic Acid on the Expression of Phase II Enzymes and the Activation of Nrf2

Based on our results, sinapic acid was selected for exploring the mechanism underlying the antihypertensive effect of phenolic acid. We measured the protein expression levels of HO-1, NQO-1, GCLC, and Nrf2. As shown in Figure 4, treatment with sinapic acid enhanced HO-1, NQO-1, and GCLC expression levels in a dose-dependent manner. In addition, sinapic acid significantly increased the nuclear translocation of Nrf2. The Nrf2 pathway is important for protection against various stressors [26]. Cytotoxicity caused by t-BHP-induced oxidative damage was recovered by caffeic acid via an increase in the expression of detoxifying enzymes, including HO-1 and GCLC [27]. Luo et al. (2018) reported that HO-1 ameliorates oxidative stress-induced endothelial aging by modulating eNOS activation [28]. Ginsenoside Rg3 upregulates the Nrf2 signaling pathway via Akt activation and improves endothelial dysfunction [29]. Moreover, sinapic acid reduces renal apoptosis, inflammation, and oxidative damage [30]. These results suggest that sinapic acid-mediated endothelial cell protection against oxidative damage may be associated with the antioxidative properties of sinapic acid.

### 3.4. Effects of Phenolic Acids on Endothelial Dysfunction

NO is essential for maintaining vascular function in the endothelium. Phosphorylation of eNOS can regulate NO production [31] and is essential for the improvement of CVD [32]. Akt mediates NO production via phosphorylation of eNOS, which promotes endothelial cell migration and angiogenesis [33]. A previous study reported that eNOS phosphorylation facilitates vasorelaxation via the PI3K/Akt signaling pathway in HUVECs [34]. Therefore, phosphorylation of eNOS and Akt is important for the treatment of endothelial dysfunction. As shown in Figure 5, treatment with H_2_O_2_ (600 μM) significantly reduced the phosphorylation of eNOS and Akt. However, sinapic acid treatment at concentrations of 12.5, 25, and 50 μM enhanced the phosphorylation of eNOS by 14.1, 26.3, and 48%, respectively, compared to that in the H_2_O_2_-treated group. Sinapic acid increased Akt phosphorylation in a dose-dependent manner. Chen et al. (2020) reported that phenolic acids extracted from ginseng protect against vascular endothelial cell injury via the activation of the PI3K/Akt/eNOS pathway [35]. Yan et al. (2020) reported that gallic acid attenuated vascular dysfunction and hypertension in angiotensin II-induced C57BL/6J mice by suppressing eNOS degradation [36]. Taken together, our results showed that sinapic acid may be effective in the treatment of endothelial dysfunction via phosphorylation of eNOS and Akt.

## 4. Conclusions

This study showed that phenolic acids significantly protected endothelial cells against H_2_O_2_-induced oxidative damage by modulating NO, GSH, MDA, and ROS levels. Sinapic acid alleviated endothelial dysfunction by enhancing HO-1, NQO-1, GCLC, p-Akt, and p-eNOS expression levels, as well as activating Nrf2 nuclear translocation. Overall, these results illustrate that sinapic acid, which exists abundantly in cereals, spices, vegetables, oil seed crops, citrus, and berry fruits, has the potential as a treatment option for hypertension. However, further in vivo studies and clinical trials are needed to determine the underlying mechanism of action.

## Figures and Tables

**Figure 1 molecules-27-06185-f001:**
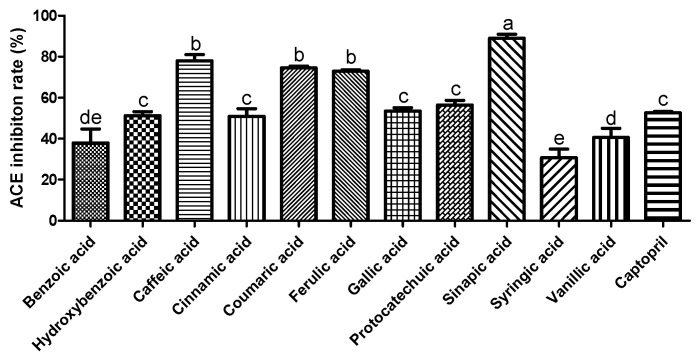
Inhibitory effect of selected phenolic acids (10 mM) on angiotensin I converting enzyme. Captopril (1.15 µM) was used as positive control. Each value was expressed as the mean ± standard error (*n* = 3). Different letters above the bars indicate significant differences based on the Duncan’s test (*p* < 0.05).

**Figure 2 molecules-27-06185-f002:**
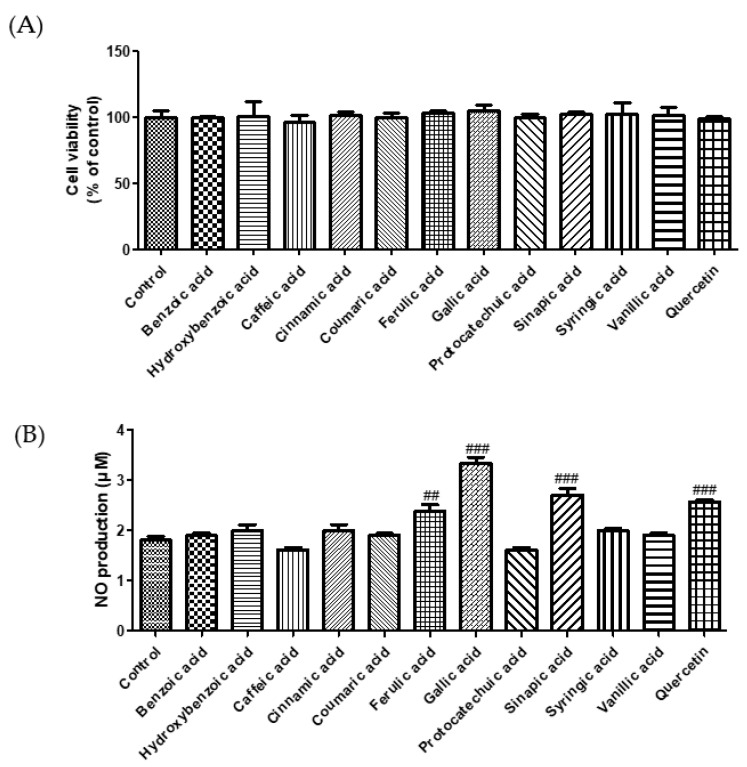
Effect of selected phenolic acids (50 μM) on cell cytotoxicity (**A**) and NO production (**B**) in EA.hy 926 cells. Quercetin (25 μM) was used as positive control. Each value was expressed as the mean ± standard error (*n* = 3). Statistical significance was analyzed using the Tukey test. ^##^
*p* < 0.01 and ^###^
*p* < 0.001 versus nontreated cells.

**Figure 3 molecules-27-06185-f003:**
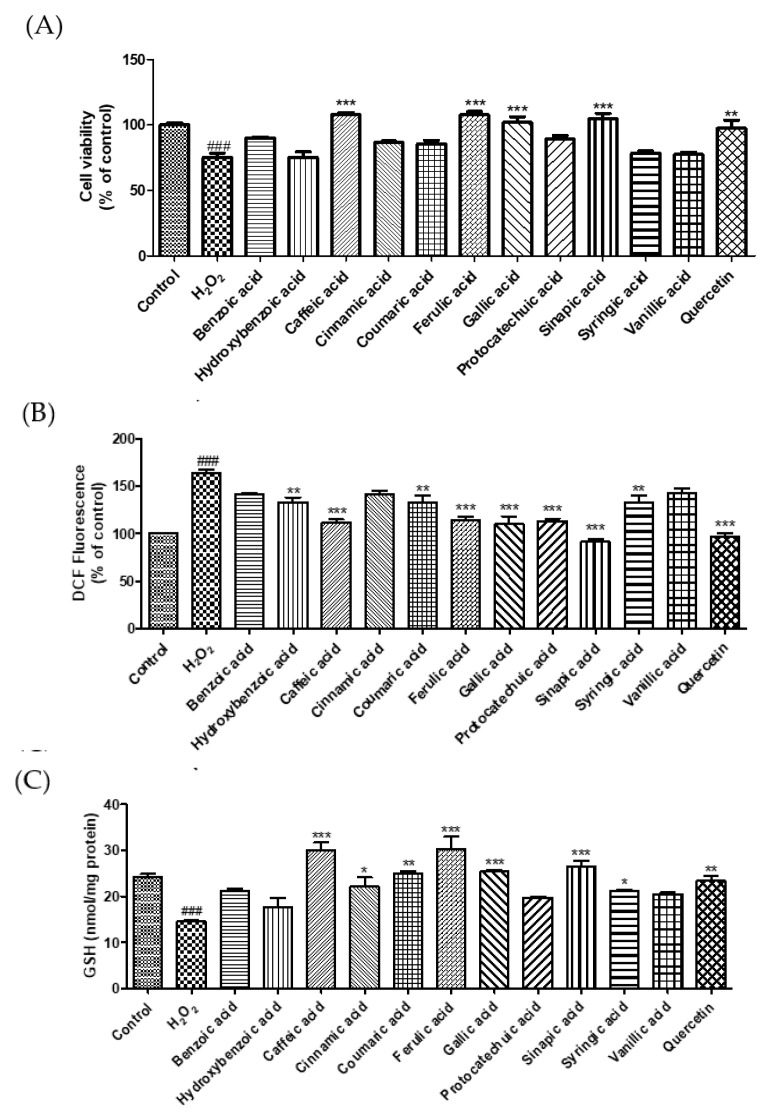
Effect of selected phenolic acids (50 μM) on cell viability (**A**), the generation of reactive oxygen species (**B**), glutathione (**C**), and malondialdehyde (**D**) against hydrogen peroxide (600 μM) in EA.hy 926 cells. Quercetin (25 μM) was used as positive control. Each value was expressed as the mean ± standard error (*n* = 3). Statistical significance was analyzed using the Tukey test. ^###^
*p* < 0.001 versus nontreated cells. * *p* < 0.05, ** *p* < 0.01, and *** *p* < 0.001 versus hydrogen-peroxide-treated cells.

**Figure 4 molecules-27-06185-f004:**
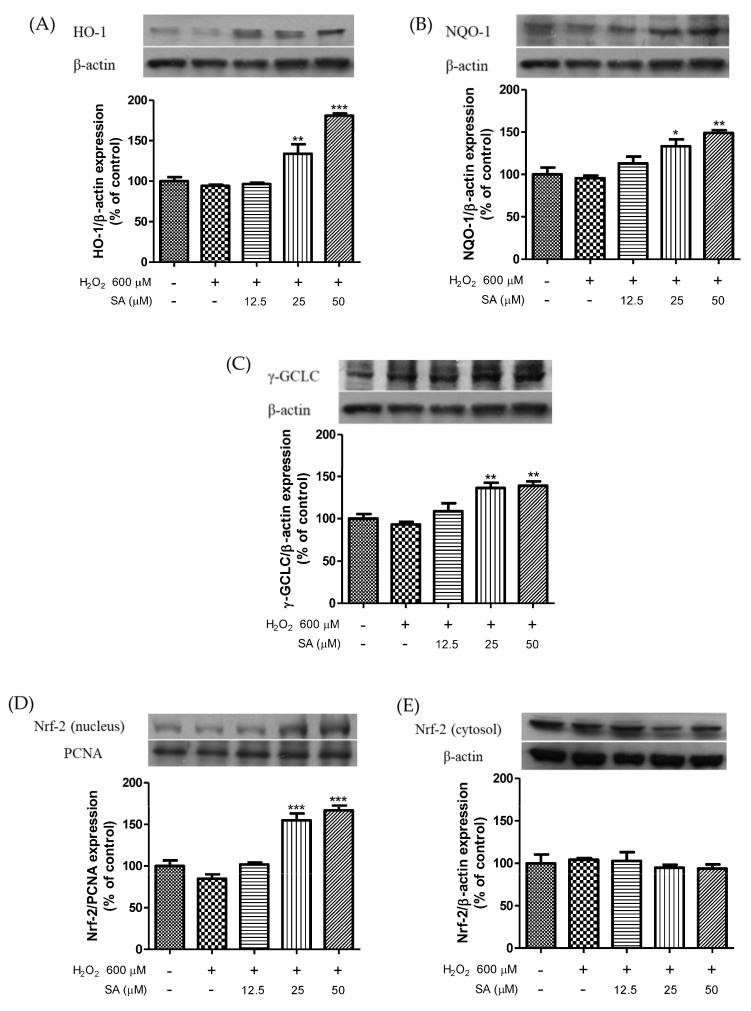
Effect of sinapic acids on HO-1 (**A**), NQO-1 (**B**), and GCLC (**C**) protein expression and Nrf-2 expression levels in nucleus (**D**) and cytosol (**E**) against hydrogen peroxide (600 μM) in EA.hy 926 cells. Each value was expressed as the mean ± standard error (*n* = 3). Statistical significance was analyzed using the Tukey test. * *p* < 0.05, ** *p* < 0.01, and *** *p* < 0.001 versus hydrogen-peroxide-treated cells.

**Figure 5 molecules-27-06185-f005:**
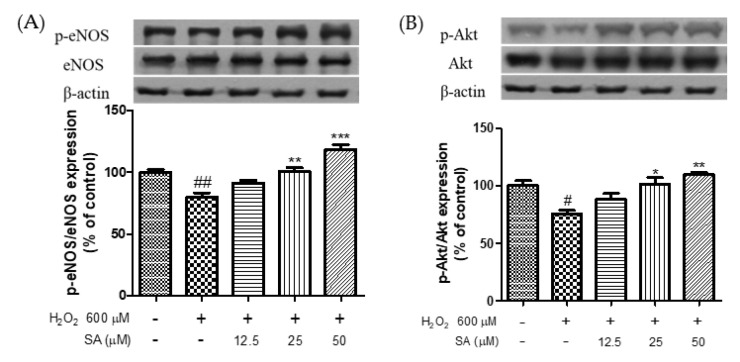
Effect of sinapic acids on p-eNOS (**A**) and p-Akt (**B**) protein expression against hydrogen peroxide (600 μM) in EA.hy 926 cells. Each value was expressed as the mean ± standard error (*n* = 3). Statistical significance was analyzed using the Tukey test. ^#^
*p* < 0.05 and ^##^
*p* < 0.01 versus nontreated cells. * *p* < 0.05, ** *p* < 0.01, and *** *p* < 0.001 versus hydrogen-peroxide-treated cells.

**Table 1 molecules-27-06185-t001:** The structure of phenolic acids.

Name	Structure	Name	Structure
Benzoic acid	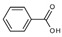	Gallic acid	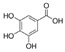
Hydroxybenzoic acid	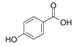	Protocatechuic acid	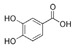
Caffeic acid	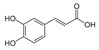	Sinapic acid	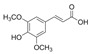
Cinnamic acid	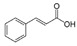	Syringic acid	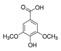
Coumaric acid	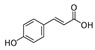	Vanillic acid	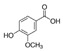
Ferulic acid	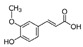	

## Data Availability

Not applicable.

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
