# Peer review of "Comparison of the Antihypertensive Activity of Phenolic Acids"

_molecules, 2022, doi:10.3390/molecules27196185_

Round 1
Reviewer 1 Report
The manuscript submitted by Yu and co-workers describes the effect of naturally occurring phenolic acids on hypertension using a cellular model with a more focus was made on sinapic.
The work is interesting and well conducted. However, modifications needs to be done before accepting its publications n Molecules.
1. It will be of interest to show in the main manuscript the chemical structures of the phenolic acids investigated in this study.
2. Figure 2. The positive control (quercetin) is used at half concentration compared to phenolic acid! What would be the activity of phenolic acid tested as 25 micromolar ? The same remark can be made for the data reported in Figure 3!
2. What is the purpose of the sentence “Flavonoids from Malus doumeri leaves…” (lines 166-167)!
3. There are some language perfections that merit to be done. For example:
- line 153 “However, treatments BY caffeic, ferulic, gallic, and …
Line 155 “To examine whether phenolic acids PROTECT endothelial cells against oxidative damage,…
Author Response
The manuscript submitted by Yu and co-workers describes the effect of naturally occurring phenolic acids on hypertension using a cellular model with a more focus was made on sinapic. The work is interesting and well conducted. However, modifications needs to be done before accepting its publications n Molecules.
1. It will be of interest to show in the main manuscript the chemical structures of the phenolic acids investigated in this study.
Response) We added the chemical structures in the main manuscript.
2. Figure 2. The positive control (quercetin) is used at half concentration compared to phenolic acid! What would be the activity of phenolic acid tested as 25 micromolar ? The same remark can be made for the data reported in Figure 3!
Response) In our preliminary experiments and published study, both concentrations of 20 uM and 50 uM of phenolic acids were compared for their anti-diabetic efficacy in HepG2 cells. However, there was no significant difference in some phenolic acids at the concentration of 20uM. The primary goal in this study was to compare the relative activities of the phenolic acids and, therefore, 50 uM of phenolic acids was used to show significant differences of many phenolic acids in activity. For positive control, quercetin, 25 uM was used because we tried to show that the efficacy of quercetin was in the middle of the activities of the phenolic acids for comparison purpose.
3. What is the purpose of the sentence “Flavonoids from Malus doumeri leaves…” (lines 166-167)!
Response) We deleted the sentence you marked and added new sentence with more relevance. Following sentence was added. “Caffeic acid and chlorogenic acid decreased blood pressure in hypertensive rat by increasing GSH and reducing MDA levels [16].”
4. There are some language perfections that merit to be done. For example:
- line 153 “However, treatments BY caffeic, ferulic, gallic, and …
Line 155 “To examine whether phenolic acids PROTECT endothelial cells against oxidative damage,…
Response) Corrected as recommended.
Reviewer 2 Report
The paper is interesting, it is not new in the sense that there are already previous studies on the antihypertensive properties of phenolic acids and plant sources or foods rich in them. However, it clearly evidences that not all phenolic compounds achieve the antihypertensive, non-cytotoxic and also antioxidant effect, which brings much more benefit with respect to other phenolic compounds, even over drugs such as captopril, so I agree with the authors' conclusions.
I have only three observations:
1. Captopril concentration (0.25 ug/mL) used is in units different from those of phenolic acids (mM) with which its effect is compared, so I would recommend to use the same units, therefore it is not clear why there is significant difference between some phenolic acids and captopril with non-comparable concentration units.
2. Why did you performed the cell treatments at concentrations of 50 uM with the phenolic acids in the assays in Figures 2 through 5, while for the assays in the results in Figure 1, the concentration of every phenolic acid including sinapinic acid was 10 mM?
3. Finally, you could mention in your conclusions or in the Introduction the food sources that contain the highest amount of the phenolic compounds studied here, mainly sinapinic acid.
Author Response
The paper is interesting, it is not new in the sense that there are already previous studies on the antihypertensive properties of phenolic acids and plant sources or foods rich in them. However, it clearly evidences that not all phenolic compounds achieve the antihypertensive, non-cytotoxic and also antioxidant effect, which brings much more benefit with respect to other phenolic compounds, even over drugs such as captopril, so I agree with the authors' conclusions.
I have only three observations:
1. Captopril concentration (0.25 ug/mL) used is in units different from those of phenolic acids (mM) with which its effect is compared, so I would recommend to use the same units, therefore it is not clear why there is significant difference between some phenolic acids and captopril with non-comparable concentration units.
Response) We revised the unit of captopril concentration from 0.25 ug/mL to 1.15 uM.
2. Why did you performed the cell treatments at concentrations of 50 uM with the phenolic acids in the assays in Figures 2 through 5, while for the assays in the results in Figure 1, the concentration of every phenolic acid including sinapinic acid was 10 mM?
Response) ACE inhibition in Fig. 1 was enzyme inhibitory assay in test tube. In this experiment, the sample concentrations can vary significantly with enzyme and substrate concentrations. In our system. 10 mM of phenolic acids was appropriate to show differences in efficacy. For figures 2 through 5, cell lines were used. In this case, the sample concentration for cell lines was decided based on cytotoxicity and efficacy of the samples.
3. Finally, you could mention in your conclusions or in the Introduction the food sources that contain the highest amount of the phenolic compounds studied here, mainly sinapinic acid.
Response) We added the sentence in conclusion. “Overall, these results illustrate that sinapic acid, which exists abundantly in cereals, spic-es, vegetables, oil seed crops, citrus, and berry fruits, has the potential as a treatment option for hypertension.”